# PAI-1 mediates acquired resistance to MET-targeted therapy in non-small cell lung cancer

Yin Min Thu[1], Ken Suzawa[1]*, Shuta Tomida[2], Kosuke Ochi[1], Shimpei Tsudaka[1], Fumiaki Takatsu[1], Keiichi Date[1], Naoki Matsuda[1], Kazuma Iwata[1], Kentaro Nakata[3], Kazuhiko Shien[1], Hiromasa Yamamoto[1], Mikio Okazaki[1], Seiichiro Sugimoto[1], Shinichi Toyooka[1]

**1** Department of General Thoracic Surgery and Breast and Endocrinological Surgery, Okayama University Graduate School of Medicine, Dentistry and Pharmaceutical Sciences, Okayama, Japan, **2** Center for Comprehensive Genomic Medicine, Okayama University Hospital, Okayama, Japan, **3** Department of Surgery, Division of Cardiovascular and Thoracic Surgery, Duke University School of Medicine, Durham, North Carolina, United States of America

* ksuzawa@okayama-u.ac.jp

**Data Availability Statement:** All relevant data are within the paper and its Supporting Information files.

## Abstract

Mechanisms underlying primary and acquired resistance to MET tyrosine kinase inhibitors (TKIs) in managing non-small cell lung cancer remain unclear. In this study, we investigated the possible mechanisms acquired for crizotinib *MET*-amplified lung carcinoma cell lines. Two *MET*-amplified lung cancer cell lines, EBC-1 and H1993, were established for acquired resistance to MET-TKI crizotinib and were functionally elucidated. Genomic and transcriptomic data were used to assess the factors contributing to the resistance mechanism, and the alterations hypothesized to confer resistance were validated. Multiple mechanisms underlie acquired resistance to crizotinib in *MET*-amplified lung cancer cell lines. In EBC-1-derived resistant cells, the overexpression of SERPINE1, the gene encoding plasminogen activator inhibitor-1 (PAI-1), mediated the drug resistance mechanism. Crizotinib resistance was addressed by combination therapy with a PAI-1 inhibitor and PAI-1 knockdown. Another mechanism of resistance in different subline cells of EBC-1 was evaluated as epithelial-to-mesenchymal transition with the upregulation of antiapoptotic proteins. In H1993-derived resistant cells, MEK inhibitors could be a potential therapeutic strategy for overcoming resistance with downstream mitogen-activated protein kinase pathway activation. In this study, we revealed the different mechanisms of acquired resistance to the *MET* inhibitor crizotinib with potential therapeutic application in patients with *MET*-amplified lung carcinoma.

## Introduction

As a leading cause of the incidence and mortality of cancer prevalence worldwide, lung cancer requires a comprehensive understanding of the nature and appropriate management of the disease, which are essential to reduce its global burden [1]. Among the incidences of lung cancer, non-small cell lung cancer (NSCLC) comprises approximately 80%–85%, which has a poor prognosis and requires complex strategies [2]. With the development of personalized

**Funding:** This work was supported by Japan Society for the Promotion of Science (JSPS) KAKENHI Grant Number JP19K18216.

**Competing interests:** I have read the journal's policy and the authors of this manuscript have the following competing interests: Shinichi Toyooka received research funding from Eli Lilly Japan, Taiho (Japan) and Chugai (Japan), and lecture fees from Chugai. All other authors have declared that no competing interests exist.

medical treatment, molecular-targeted therapies have been used in various types of cancers [2]. *MET* proto-oncogene alteration, including the exon 14 skipping mutation (*MET*ex14) and amplification, comprises about 6.5% of lung adenocarcinoma, and this alteration is the driver oncogene [3]. *MET*ex14 is the common alteration of *MET*, which is often mutually exclusive with other oncogenic drivers (*EGFR*, *KRAS*, or *ALK*). At present, some MET tyrosine kinase inhibitors (TKIs) have been demonstrated to be highly effective in this molecular subgroup of patients [4–6]. *MET* amplification comprises 1.7% of all early and 2.5% of metastatic lung adenocarcinomas, and high-level *MET* amplification is thought to be an oncogenic driver [7–9].

The *MET* gene encodes the MET tyrosine kinase receptor or hepatocyte growth factor (HGF) receptor, and several mechanisms are involved in the activation of the MET signaling pathway [10]. MET is activated when the HGF ligand binds to the MET receptor, which induces the homodimerization and phosphorylation of Y1234 and Y1235 tyrosine kinase residues and regulates tyrosine kinase activity. The carboxy-terminal tail harbors Y1349 and Y1356 tyrosine kinase residues that serve as docking sites upon phosphorylation of intracellular adaptor proteins [11]. Downstream signaling pathways for MET include mitogen-activated protein kinase (MAPK) cascades such as extracellular signal-regulated kinase 1 (ERK1) and ERK2, p38 and Jun amino-terminal kinases (JNKs), phosphoinositide 1-kinase-AKT (PI3K-AKT) signaling axis, signal transducer and activator of transcription proteins, and nuclear factor-κb (NFκb)-NFκb inhibitor-α (iκbα) complex [10]. The abnormal activation of the MET/HGF pathway caused by *MET*ex14 or MET amplification is involved in the maintenance of tumor transformation and promotion of cancer cell proliferation, survival, invasion, and metastasis [12,13]. Therefore, the MET signaling pathway can be targeted by several mechanisms, including selective TKIs or multi-target TKIs, antibodies against MET, or its HGF ligand [14–16].

Among the therapeutic targets against MET, crizotinib (PF-02341066) is a multikinase inhibitor targeting ROS1, ALK, and MET. This inhibitor competes with the MET tyrosine kinase domain at the adenosine triphosphate binding site and prevents further activation of the MET receptor and its downstream signal transduction [12,17]. According to the National Comprehensive Cancer Network guidelines, crizotinib has been recommended for targeted therapy against high-level *MET* amplification or *MET* exon 14 skipping mutation based on the results of clinical trials [4,8,18]. However, similar to most kinase-driven cancers, acquired resistance to MET-targeted therapy eventually arises by various mechanisms, including on-target resistance in kinase domain and off-target resistance by bypass signaling pathway activation, copy number alteration, and histologic transformation [19,20]. In this study, we established crizotinib-resistant *MET*-amplified NSCLC cell lines and elucidated their underlying resistance mechanisms to obtain comprehensive insights into this issue. We demonstrated several mechanisms of crizotinib resistance in distinct cell lines using different methods, including genomic and in vitro assay analyses.

## Materials and methods

### Cell lines and reagents

Two *MET*-amplified cell lines, namely, EBC-1 (*MET*-amplified lung squamous cell carcinoma) and NCI-H1993 (*MET*-amplified adenocarcinoma), were purchased from RIKEN Cell Bank, Tsukuba, Japan and ATCC, American Type Culture Collection (Manassas, Virginia), respectively. All cells were cultured in RPMI-1640 medium supplemented with 10% FBS under a culture condition in a 5% $CO_2$-supplied humidified incubator at 37°C. Crizotinib-resistant cell lines were established using two methods: high-concentration exposure method using crizotinib 1 μM from the start of culture or stepwise escalation method using the concentration of

crizotinib from 0.1 to 1 μM to the parental cells. Crizotinib stepwise resistant cell lines (CRS) were established by treating the cells with increasing doses of crizotinib (from 0.1 μM to 1 μM) for 6 months. The cells were first exposed to crizotinib 0.1μM until they were damaged at around 30% confluence. They were then passaged to reach about 80% confluence in a drug-free state. When these cells reached 80% confluence, the drug was exposed again. The cells were repeatedly treated with the same concentration of crizotinib until almost all the cells survived the treatment. When the cells survived the treatment, the drug exposure was increased with a higher dose of 0.2 μM. This process was repeated with a stepwise dose of crizotinib until the cells survived the 1 μM concentration. In the high-concentration exposure method establishment for crizotinib resistant high-dose (CRH) cell lines, the cells were exposed directly to the high dose of 1000nM crizotinib from the start until about 30% confluence of the cells were left after exposure. We then passaged them until they reached the 80% confluence in a drug-free state in a similar manner. We then treat them again with the 1 μM crizotinib until almost all of the cells have survived the treatment. Finally, the resistant cell lines were designated as EBC-1 CRS and H1993 CRS for the crizotinib-resistant stepwise method and EBC-1 CRH and H1993 CRH for the crizotinib-resistant high-concentration exposure method. Crizotinib (PF-02341066), cabozantinib (XL184), trametinib (GSK1120212), and tiplaxtinin (PAI0-039) were purchased from Selleck Chemicals (Houston TX, USA), and tepotinib (CAS1103508-80-0) was purchased from TargetMol (Massachusetts, United States).

## Cell viability assays

Cells were cultured at 2000 cells per well in 96-well plates and treated with drugs for 72 h. Cell viability was then determined by using the modified MTS assay with CellTitre 96 Aqueous One Solution Reagent (Promega, Madison, WI United States). Each experiment was performed in six-replicates in three independent experiments and $IC_{50}$ values were present at drug concentration μM ± SD (standard deviation).

## Copy number analysis

Genomic DNA extraction from cell lines was done by using DNEasy Blood and Tissue kit (Qiagen, Venlo, Netherlands). Copy number gains (CNGs) of *MET* gene (forward frimer: "ATCAACATGGCTCTAGTTGTC" and reverse primer "GGGAGAATATGCAGTGAACC") was determined by using quantitative real-time (q-RT) PCR, performed on StepOnePlus Real-Time PCR System (Thermofisher Scientific, Waltham, MA, USA). Power SYBR Green PCR Master Mix (Thermofisher Scientific, Waltham, MA, USA) was used, and *LINE-1* gene (forward primer: "AAAGCCGCTCAACTACATGG" and reverse primer: "TGCTTTGAATGCGT CCCAGAG") was used as a reference gene. The samples were analyzed in three replicate data and the relative copy number of each sample was determined by comparing it with Human Genomic DNA (Promega, Madison, WI United States).

## Combination assay

For the two-drug combinations, cells were cultured at 2000 cells per well in 96-well plates and treated with different concentrations of crizotinib and trametinib for 72 h. Cell viability was measured using the MTS assay, and the inhibition effect was calculated. Using the Chou–Talalay method [21]., Compusyn (https://www.combosyn.com/, Combosyn Inc., Paramus, NJ, USA) was used to evaluate synergism. The combination index (CI) and fraction affected (Fa) were calculated using nonconstant ratio drug combination analysis in accordance with the software instructions, and CI plots were generated. Drug synergism, additive effect, and antagonism were defined by CI values of $<1$, $= 1$, and $>1$, respectively.

## Western blot analysis

Cells were washed in PBS at 4°C, and the total cell lysate was extracted using lysis buffer, which is a mixture of RIPA buffer, phosphate inhibitor cocktails 2 and 3 (Sigma-Aldrich, Burlington, MA, United States), and complete mini protease inhibitor cocktail (Roche, Basel, Switzerland). The primary antibodies used for Western blot analysis were MET, phospho-MET, EGFR, phospho-EFGR, AKT, phospho-AKT, ERK, phospho-ERK, PAI-1, E-cadherin, beta-catenin, N-cadherin, vimentin, snail, slug, BCL-2 (B-cell lymphoma 2), poly ADP-ribose polymerase (PARP), cleaved PARP, and glyceraldehyde-3-phosphate dehydrogenase (GAPDH) from Cell Signaling Technology, Danvers, MA, USA), and Zeb1 (Santa Cruz Biotechnology, Texas, USA). The secondary antibodies used were HRP-conjugated anti-rabbit or anti-mouse IgG (Cell Signaling Technology, Danvers, MA, USA). Specific signals were detected using the ECL Prime Western Blotting Detection System (Cytiva, Amersham, UK) and a LAS-4000 Luminescent Image Analyzer (Fujifilm, Tokyo, Japan). Phosphorylation and apoptosis were examined after drug exposure of cells for 2 and 48 h, respectively. All original uncropped western blots are shown in S1 Fig. The relative band intensities were quantified using ImageJ software (National Institute of Health).

## Colony formation assay

The cells were cultured at 500 cells per well in six-well plates, treated with DMSO (Control) and desired drugs, individually or in combination, for 14 days, fixed in 4% formaldehyde, and stained with 0.2% crystal violet (Sysmex, Hyogo, Japan). The colony counts were determined by using ImageJ. Each experiment was performed in three replicated independent experiments. The relative number of colonies formed was compared.

## RNA sequencing

Total RNA extraction was performed using the RNeasy Mini Kit (Qiagen, Venlo, Netherlands), and RNA samples were quantified using a Qubit 2.0 Fluorometer (ThermoFisher Scientific, Waltham, MA, USA). In addition, RNA integrity was checked using an Agilent TapeStation (Agilent Technologies, Palo Alto, CA, USA). RNA sequencing libraries were prepared using the TruSeq RNA Access Library Prep Kit (Illumina, San Diego, USA) according to the manufacturer's instructions. The library samples were sent to Azenta Life Sciences (Tokyo, Japan) for RNA-seq analysis, and RNA sequencing was performed on the Illumina HiSeq 2500 System (Illumina) in a 2 × 150 bp paired-end sequencing protocol. All sequence reads were converted to FASTQ format using Bcl2Fastq. CLC Genomics Workbench 20.0.2 (Qiagen) was used to count the reads mapped onto each gene via the RNA-seq analysis pipeline. The human genome (hg38) (https://www.ncbi.nlm.nih.gov/datasets/genome/GCF_000001405.37/) was used as the reference sequence. Transcripts per million were calculated for further normalization to account for sample variations.

## Transcriptomic bioinformatics analysis

Gene set enrichment analysis (GSEA) was performed using GSEA v4.1.0 software downloaded from the GSEA website (http://software.broadinstitute.org/gsea/index.jsp). Gene sets were obtained from the molecular signature database (MSigDB) v5.0 (https://www.gsea-msigdb.org/gsea/). The parental and resistant cell lines were analyzed by supervised hierarchical clustering of 189 genes with significant differences using the Cluster 3.0 program (http://bonsai.hgc.jp/~mdehoon/software/cluster/software.htm#ctv). An average linkage was performed for clustering, and the Java Tree View Program was used to generate a heatmap of the clustering

result (https://jtreeview.sourceforge.net). Functional annotation analysis of the database for annotation, visualization, and integrated discovery (DAVID) was performed using DAVID Web version 6.8 (https://david.ncifcrf.gov/) for the classification of gene ontology (GO) terms in cellular component (CC), biological process (BP), and molecular function (MF).

## mRNA gene expression analysis

Total RNA extraction was performed using the RNeasy Mini Kit (Qiagen). Total RNA was then converted into complementary DNA (cDNA) using a high-capacity cDNA Reverse transcription kit (Thermofisher Scientific). The cDNA was used for qRT-PCR analysis using the TaqMan Gene Expression Assay or Power SYBR Green PCR Master Mix (Applied Biosystems) and the ABI StepOnePlus Real-Time PCR instrument (Thermofisher Scientific). The mRNA expression of PAI-1 (Taqman Assay Hs00167155_m1) was calculated using the $\Delta\Delta$CT method. The GAPDH gene (Taqman Assay Hs02786624_g1) was used as the endogenous control. The samples were analyzed in three replicates for each experiment.

## Establishment of knockdown cell lines

Knockdown cell lines were established using short hairpin RNA (shRNA) targeted to *SERPINE1* (#SHCLNG-NM_000602), and the control shRNA nontarget plasmid was purchased from Sigma-Aldrich (Merck). The information on the target sequence was as follows: shPAI-1#1, CCTCATCCACAGCTGTCATAG (#TRCN0000370107); shPAI-1#2, TCTCTGCCCTCACC AACATTC (#TRCN0000331004); and a nontarget shRNA, with no known gene targets from any species. The plasmid was prepared using the HiSpeed Plasmid Midi kit (Qiagen). Virus production was performed in Lenti 293T cells using pLKO-1-puro as a vector, along with packaging vectors (pVSVG and psPAX2) cultured at $3.0 \times 10^5$ cells per well in six-well plates. The cDNA of shPAI-1#1 and shPAI-1#2 and the nontarget control pLKO-1-puro were added at 1 µg/mL. The supernatant viral media was obtained 48 h after transfection and added to EBC-1 CRH cells. The stable cell lines of sh*SERPINE1* knockdown were treated with 4 µg/mL of puromycin twice for 48 h with a two-day interval.

## Survival analysis using Kaplan-Meier plotter

We performed the survival analysis of SERPINE1 gene by using web based data analysis Kaplan-Meier (www.kmplot.com). The lung cancer studies were selected from the database and SERPINE1 (Affy ID: 202628_s_at) was used to perform the overall survival multivariate analysis among all lung cancer patients (n = 2166).

## Statistical analysis

Statistical analyses were performed using GraphPad Prism, version 9.2.0 (GraphPad Software, San Diego, CA, USA). All group differences were compared using the t-test for repeated measurements. A *p*-value $< 0.05$ was considered statistically significant.

# Results

## Establishment of crizotinib-resistant cell lines

Two types of crizotinib-resistant cell lines were established in two parental cell lines with *MET* amplification, EBC-1 and H1993, using two different methods: the high-dose concentration method and the stepwise escalation method. Cell viability assays were performed to confirm crizotinib resistance. The IC$_{50}$ value of crizotinib in EBC-1 parental, EBC-1 CRH, and EBC-1 CRS were 0.043 $\pm$ 0.05, 2.115 $\pm$ 0.04 and 1.731 $\pm$ 0.11 µM $\pm$ SD, respectively. The IC$_{50}$ value of

H1993 parental, H1993 CRH, and H1993 CRS were 0.283 ± 0.13, 3.564 ± 0.08 and 4.376 ± 0.05 μM ± SD, respectively. We also found that the crizotinib-resistant cell lines conferred cross-resistance to tepotinib, and the $IC_{50}$ values were 0.006 ± 0.01, 8.062 ± 0.00 and 7.635 ± 0.06 μM ± SD in EBC-1 parental, CRH, and CRS cells, respectively, and 0.065 ± 0.09, 9.375 ± NA and 5.05 ± 0.07 μM ± SD in H1933 parental, CRS, and CRH cells, respectively. Similarly, the $IC_{50}$ value of cabozantinib was 0.089 ± 0.06, 3.618 ± 0.04 and 9.09 ± NA μM ± SD in EBC-1 parental, CRH, and CRS; 2.179 ± 0.09, 13 ± 0.05 and 10.31 ± 0.06 in H1993 parental, CRH and CRS μM ± SD respectively (Fig 1A). Compared with parental cells, morphological changes in resistant cells could be observed on examination under a light microscope. For EBC-1 CRH, EBC-1 CRS, and H1993 CRH cell lines, a gain of spindle-shaped formation and elongated structure was detected, suggesting the acquisition of epithelial-to-mesenchymal transition (EMT) features (Fig 1B). Since both the cell lines were known to be *MET* amplified and to determine any intrinsic changes underlying *MET* gene amplification, we performed the copy number assay analysis in all the parental and resistant cell lines. All of the parental and resistant cell lines showed higher copy number gain and the resistant cell lines retained the property of *MET* gene amplification (Fig 1C).

The protein expression profile of the cells was analyzed by Western blotting. The key factors involved in the MET signaling pathway, including other downstream markers of cancer signaling, were examined. We found that the strong MET expression reserved in crizotinib-resistant cell lines with phospho-MET (p-MET) was suppressed in both parental and resistance cell lines with crizotinib treatment. Phospho-EGFR (p-EGFR) was also suppressed in both parental and resistant clones. Downstream AKT has slightly inhibited in EBC-1 resistant clones and completely in H1993 resistant clones. Different expression features in each parental and resistant cell were detected, with remarkably increased expression in the phosphorylated form for ERK (p-ERK) activation in EBC-1 resistant cell lines, and H1993 CRH cells were observed when treated with crizotinib (Fig 1D). Next, transcriptomic data of the respective cell lines were analyzed by RNA sequence analysis, and hierarchical clustering analysis was performed among the highly expressed significant 189 genes, which showed a significant difference with a *p*-value less than 0.05 in parental cells compared with resistant cell lines (Fig 1E). In both cell lines, the gene expression profiles of parental and resistant cells differed greatly. In addition, the different methods for establishing resistant cells (CRH and CRS, as described above) led to distinct expression profiles.

## Overexpression of PAI-1 promotes resistance to crizotinib in EBC-1 CRH cells

Based on transcriptomic RNA sequence data, the gene *SERPINE1* (Serpin Family E Member 1), which encodes the protein plasminogen activator inhibitor-1 (PAI-1), was found to be significantly expressed in EBC-1 CRH cells, which is nearly six times the expression in parental cells (Fig 2A). In confirming its expression, a Western blot on the expression of PAI-1 was also assessed, and the result showed that PAI-1 was highly expressed in EBC-1 CRH, which was consistent with gene expression data (Fig 2B). PAI-1 is a fibrinolysis-regulating protein. Furthermore, in the field of cancer, PAI-1 has been reported to be associated with tumor progression, metastasis, and angiogenesis [22]. Thus, we investigated the potential involvement of PAI-1 in resistance to MET-TKI. First, we examined the effect of using the PAI-1 inhibitor tiplaxtinin in combination treatment with crizotinib. The MTS assay was performed on EBC-1 CRH cell lines by combined treatment with tiplaxtinin 10 μM, and cell viability was checked by crizotinib treatment. The $IC_{50}$ value of crizotinib was reduced from 2.115 ± 0.09 μM ± SD in single therapy to 1.018 ± 0.07 μM ± SD when combined with tiplaxtinin. (Fig 2C). A colony

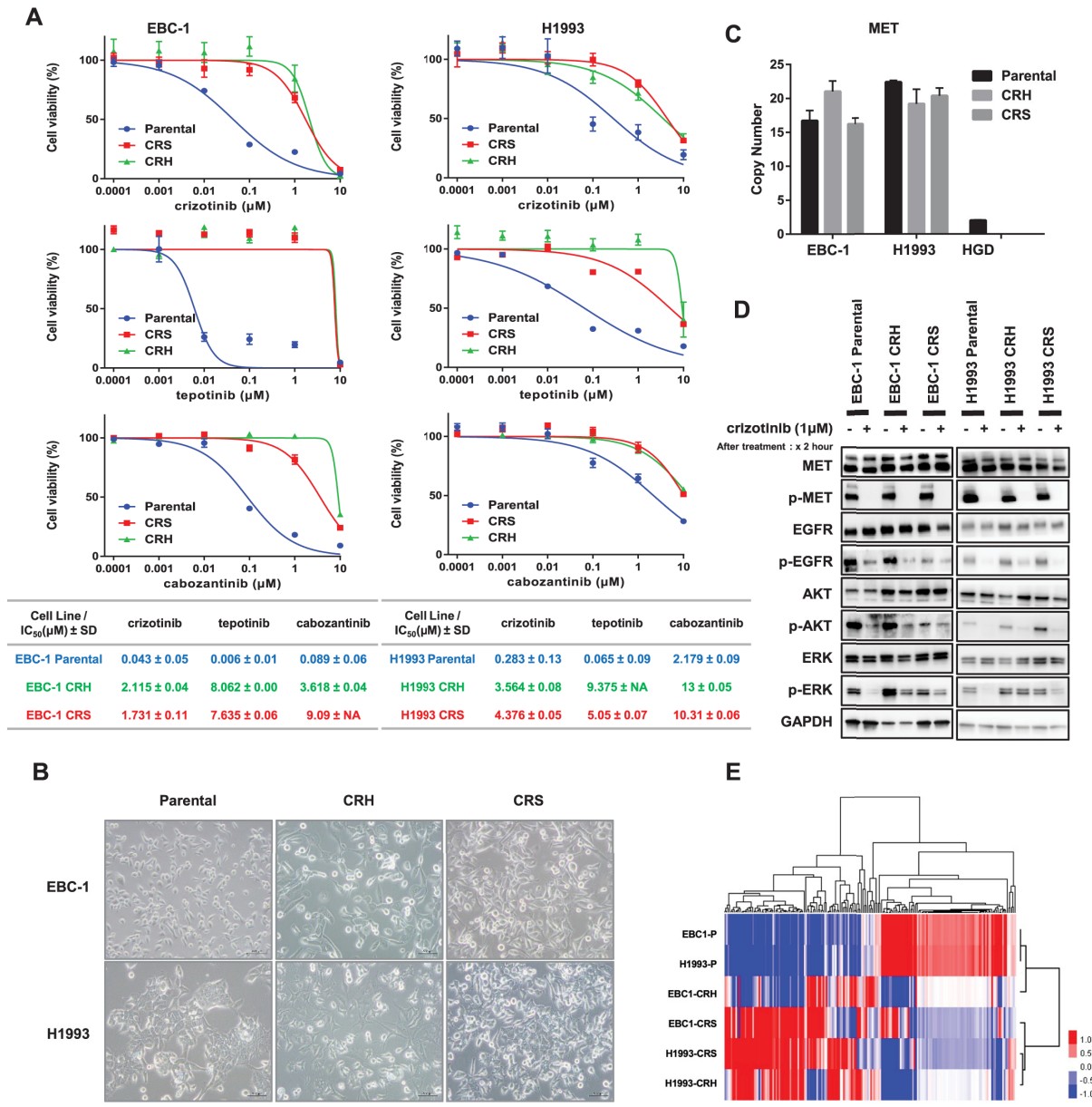

**Fig 1. Establishment of crizotinib-resistant clones.** (A) EBC-1, H1993 parental, and resistant cells were treated with crizotinib, tepotinib, and cabozantinib for 72 h, and cell viability was measured using an MTS assay. Each experiment was assayed in six replicates, and the average $IC_{50}$ values (μM ± SD) were calculated from three independent experiments. (B) Morphological changes in parental and resistant cell lines were observed under a light microscope, and features were noted (40× magnification, light microscope). (C) Copy number assay on *MET* gene. The *MET* gene copy number was determined in human genomic DNA (HGD) and EBC-1 and H1993 parental and resistant cell lines. The copy number of each cell line with relative to the HGD copy number of 2 was presented, using the *LINE-1* gene as endogenous control. (D) Protein expression analysis by Western blot in all parental and resistance cell lines. MET downstream signaling pathway markers were identified as any activation with or without crizotinib treatment. The cells were treated with crizotinib 1 μM for 2 h and subjected to immunoblotting. (E) Supervised hierarchical clustering analysis among the highly expressed significant 189 genes, which showed a significant difference with a *p*-value of less than 0.05 in parental cell lines compared with resistant cell lines. p- (Phosphorylated-), SD–standard deviation, NA–not applicable.

formation assay was performed to evaluate the survival of EBC-1 CRH cells after either individual treatment or combination treatment with crizotinib and tiplaxtinin. Colony formation in combined treatment was significantly reduced compared with individual treatments of crizotinib alone ($p = 0.011$), tiplaxtinin alone ($p = 0.013$), or without drugs ($p < 0.001$; Fig 2D).

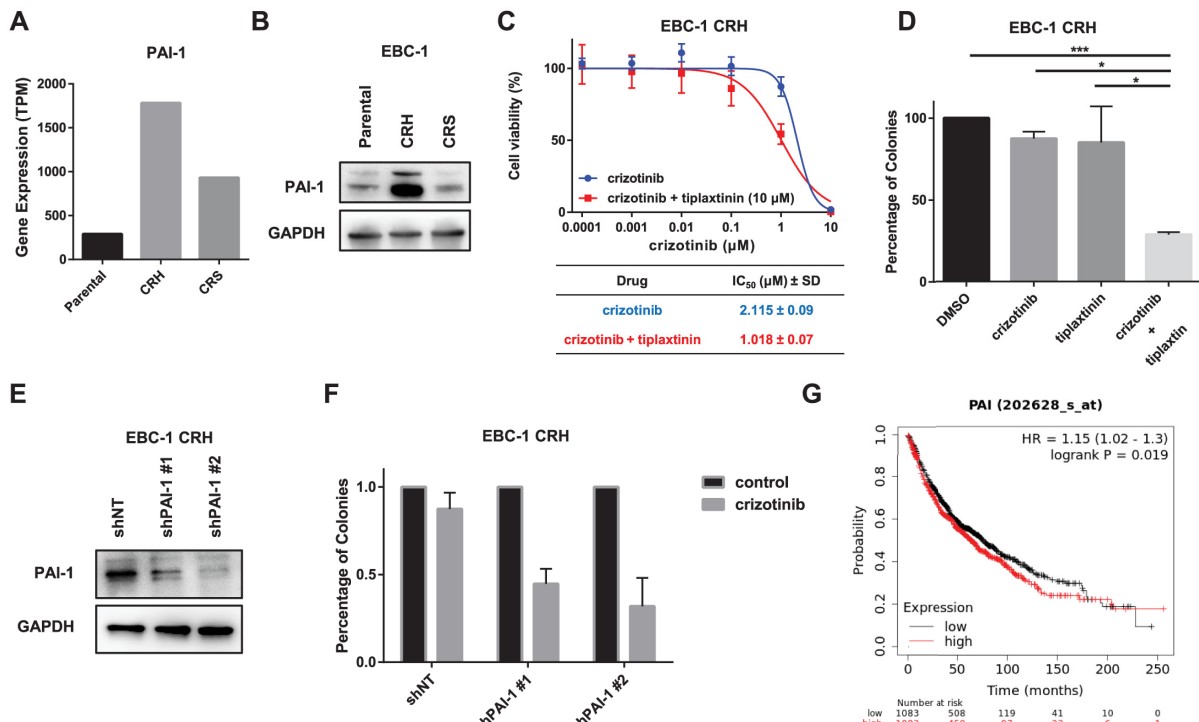

**Fig 2. Contribution of PAI-1 upregulation in the EBC-1 CRH cell line.** (A) Expression of PAI-1 (*SERPINE1* gene) in the EBC-1 cell line by RNA sequence analysis. (B) Protein expression analysis by Western blot of PAI-1 on EBC-1 cell lines showing consistent expression with RNA sequence analysis. (C) EBC-1 CRH cells were treated with crizotinib and the PAI-1 inhibitor tiplaxtinin (10μM) for 72 h, and cell viability was measured using the MTS assay. Each experiment was assayed in six-replicates, and the average $IC_{50}$ value (μM ± SD) was calculated from three independent experiments. (D) Combination treatment with crizotinib and tiplaxtinin inhibited colony formation in EBC-1 CRH cells. The cells were treated with DMSO (control), crizotinib 0.2 μM, and tiplaxtinin 10 μM, individually or in combination. The relative number of colonies formed was compared to evaluate the effect of therapeutic agents. The average of three independent experiments was measured. (E) Knockdown of PAI-1 by transfection with shRNA. Control cell lines were treated with shNT (nontarget), and two knockdown cell lines were established and validated by assessing the expression of PAI-1 by Western blot. (F) Colony formation assay analysis of EBC-1 CRH PAI-1 knockdown and control cell lines treated with DMSO (control) or crizotinib (0.2 μM) in a similar manner. (G) Kaplan–Meier analysis of the overall survival (OS) of patient with high and low expression of *SERPINE1* gene in lung cancer (HR = 1.15, longrank *P* = 0.019). *$p < 0.05$, **$p < 0.01$, ***$p < 0.00$.

Next, a knockdown experiment was performed using two independent shRNAs to assess the effect of PAI-1 overexpression on resistance. The expression of PAI-1 mRNA and protein was suppressed by 80%–90% in knockdown cell lines compared with negative control nontarget cells (Figs 2E and S2). Colony formation was reduced in shPAI-1 cells treated with crizotinib compared with the control, indicating that crizotinib resistance was addressed by the knockdown of PAI-1 (Fig 2F).

We then assessed the prognostic value of PAI-1 in patients with NSCLC (n = 2166) using the Kaplan–Meier plotter database. Patients with high expression of PAI-1 were found to be associated with poor overall survival (OS; *p* <0.05, Fig 2G). These findings suggested that the overexpression of PAI-1 promotes tumor progression and contributes to crizotinib resistance.

## MAPK pathway activation as a resistant mechanism in EBC-1 CRS cell line

Since the MET downstream marker p-ERK was upregulated in the EBC-1 crizotinib-resistant cell lines (Fig 1D), this activation was investigated by using MEK inhibitor trametinib treatment. The EBC-1 CRS cells were exposed to either crizotinib or trametinib or in combination for 2 h and protein expression analysis was performed on the downstream MET pathway

markers. The phosphorylation of ERK was found to be inhibited by the crizotinib and trameti-nib combination treatment (Fig 3A). To validate this finding on the apoptotic activity induction, the expression of cleaved PARP was analyzed after treating the cells with crizotinib and trametinib alone or in combination treatment. The cleaved PARP expression was reduced during crizotinib treatment due to resistance, but increased in the trametinib alone and more increased in combination treatment (Fig 3B), indicating that the MEK inhibitor combination treatment is effective in EBC-1 CRS cells. For further insight into these results, we also explored the sensitivity to combination treatment by using cell viability MTS assay. The IC$_{50}$ of crizotinib in the EBC-1 CRH cells reduced from 2.487 ± 0.06 μM ± SD in single therapy to 0.453 ± 0.13 μM ± SD in combination with trametinib 100 nM combination therapy (Fig 3C). We further explored the combinational effect between crizotinib and trametinib. EBC-1 CRS cells were treated with several different concentrations of crizotinib and trametinib, and growth inhibition percentage was calculated. We then analyzed the effect of combination of these two drugs by using the Chou–Talalay method for drug combination (Fig 3D). The resulting CI is defined for the additive effect (CI = 1), synergism (CI < 1), and antagonism (CI > 1) in drug combinations. In EBC-1 CRS, the CI value of the combination points for crizotinib and trametinib was less than 1, indicating that the combination of the drugs showed a synergistic effect.

### Enrichment of EMT-related gene sets in the EBC-1 CRS cell line

As mentioned previously, the microscopic features of the EBC-1 CRS cell line acquired EMT features such as spindle-shaped formation, cell elongation, and cell individualization, which could be observed under a light microscope (Fig 1B). The RNA sequence transcriptomic data were analyzed, and GSEA using hallmark gene sets showed the enrichment of EMT-related genes in the EBC-1 CRS cell line with an enrichment score (ES) of 0.52, nominal *p*-value of 0.0, and FDR q-value of 0.0 (Fig 3E). EMT-related gene sets were enriched not only in the hallmark gene sets but also in curated gene sets with a nominal *p*-value of less than 0.01 and an FDR q-value of less than 5% (S3 Fig). To confirm this result, the expression of EMT-related genes was analyzed. The results showed that the epithelial marker gene *CDH1* encoding E-cadherin was slightly reduced, and the mesenchymal marker genes *CDH2* encoding N-cadherin, *VIM* encoding vimentin, and *SNAI2* encoding Slug were upregulated in EBC-1 CRS by 1.52-, 2.39-, and 3.57-fold compared with the parental cell line (Fig 3F). This finding was further validated by protein expression analysis of EMT-related proteins by western blot analysis, where the protein expression profile was found to be in concordance with the gene expression result. The epithelial marker, beta-catenin, was downregulated and mesenchymal markers, N-cadherin and Slug, were observed to be upregulated in in EBC-1 CRS compared with the parental cell line (Fig 3G).

For further exploration of gene signatures in the EBC-1 CRS cell line, we applied a different bioinformatics approach using DAVID. First, genes whose expression level in the log2 ratio of EBC-1 CRS to parental cells was more than or equal to 2 were filtered. Among the 466 genes that were highly expressed in EBC-1 CRS, functional annotation chart analysis was performed using DAVID. Among the GO terms, analysis of CC (Cellular Component), extracellular matrix, space, and region-related genes was enriched at the top terms with *p*-values less than 0.0001 (Fig 3H). Apart from CC, the GO terms of BP (Biological Processes) and MF (Molecular function) were found to be associated with extracellular matrix structure and organization, and matrix metalloproteinases greatly contributed to EMT (S4A and S4B Fig).

Furthermore, in the EBC-1 CRS cell line, the expression level of the antiapoptotic protein Bcl-2 was upregulated (Fig 3I). Considering that Bcl-2 is an antiapoptotic member of the Bcl-2

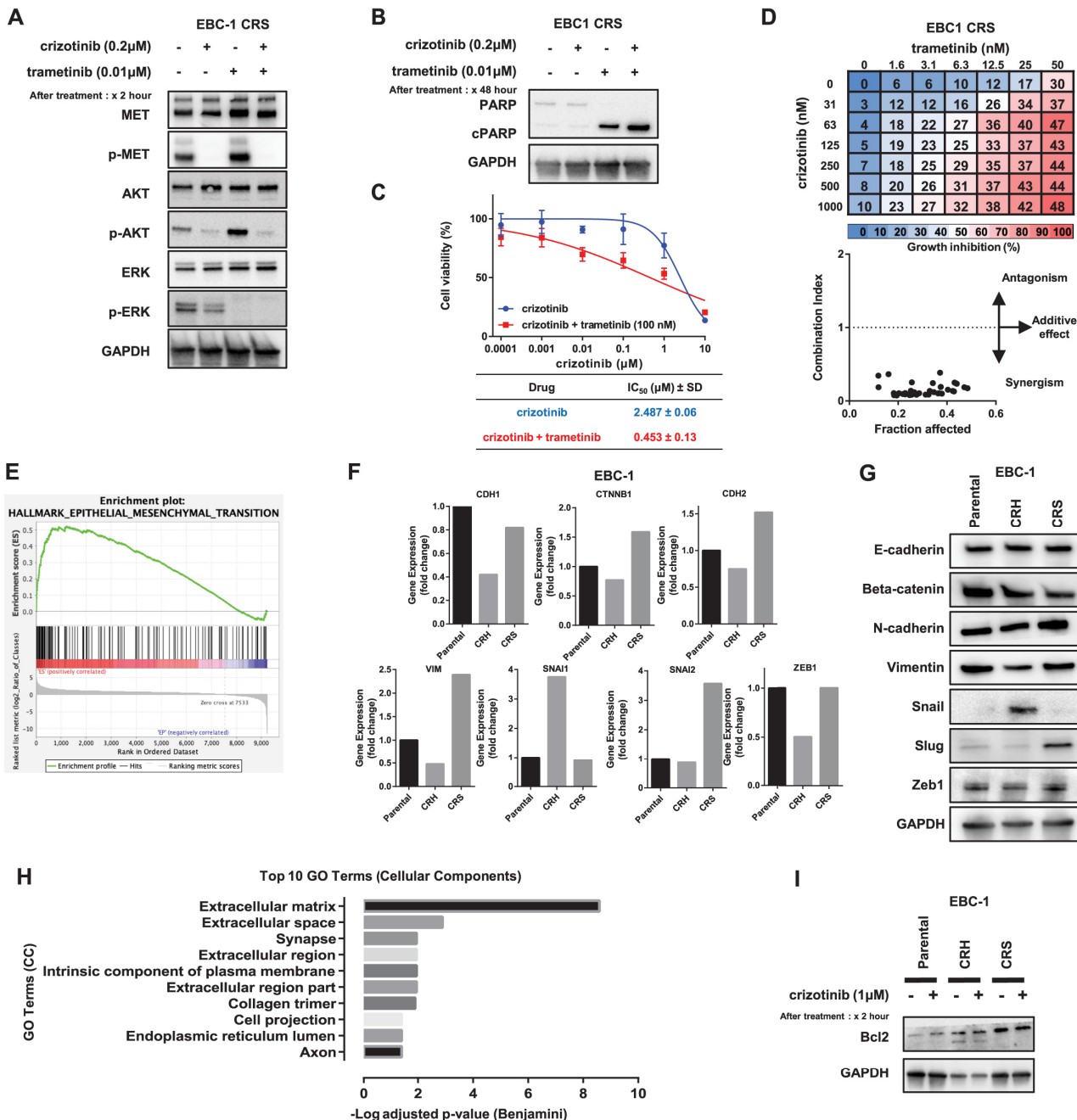

**Fig 3. MAPK pathway activation and EMT as a resistant mechanism in the EBC-1 CRS cell line.** (A) EBC-1 CRS cells were treated with crizotinib (0.2 μM) and trametinib (0.1 μM) for 2 h, individually and in combination, and the expression of MET downstream signaling markers was detected by immunoblotting. (B) The same cell line was treated with individual or a combination of crizotinib (0.2 μM) and trametinib (0.1 μM) for 48 h, and the lysate was subjected to immunoblotting on the apoptosis activity expression of PARP and cleaved PARP. (C) EBC-1 CRS cells were treated with crizotinib and trametinib (100nM) for 72 h, and the cell viability was measured using MTS assay. Each experiment was assayed in six-replicates, and the average $IC_{50}$ value (μM ± SD) was calculated from three independent experiments. (D) EBC-1 CRS cells were treated with a combination of crizotinib and trametinib for 96 h, and cell viability was determined. The mean value of the growth inhibition percent at each concentration was presented. The combination index (CI) plot indicates the CI and fraction affected (inhibition ratio) among the different drug concentrations. (E) Gene set enrichment analysis (GSEA) of the EBC-1 CRS cell line compared with parental cells in which the enrichment plot of the hallmark EMT-related gene set was presented, with an enrichment score (ES) of 0.52 and a nominal *p*-value of 0.0. (F) Gene expression data of RNA sequence analysis on the EMT-related genes *CDH1*, *CTNN2B*, *CDH2*, *VIM*, *SNAI1*, *SNAI2*, and *ZEB1* encoding E-cadherin, beta-catenin, N-cadherin, vimentin, Snail, Slug, and Zeb1, respectively. Fold changes in each resistant cell line were calculated and compared with those in the parental cell line. (G) Western blot analysis of EMT-related protein expression in EBC-1 cell lines,

with upregulated mesenchymal markers as well as N-cadherin and slug in EBC-1 CRS cell lines compared with parental cells. (H) DAVID functional gene ontology (GO) analysis of gene enrichment in the EBC-1 CRS cell line. The top 10 terms on the cellular component (CC) gene set were presented, and enrichment of genes related to the extracellular matrix was observed. (I) The expression of the antiapoptotic protein Bcl-2 in the EBC-1 CRS cell line was determined after treatment with or without crizotinib (1 μM) for 2 h and subjected to immunoblotting.

family, it modulates the apoptotic pathway, conferring a survival advantage to the cells by protecting them from apoptotic death.

## MAPK pathway activation and sensitivity to MEK inhibitors in H1993 crizotinib-resistant cells

The protein expression of parental and crizotinib-resistant cells was determined using Western blotting, and H1993 CRH cells expressed phosphor-ERK activation in the presence of crizotinib treatment (Fig 1D). Such protein expression was further investigated by combination treatment with the MEK inhibitor trametinib. The crizotinib-resistant H1993 CRH cells were treated with either crizotinib or trametinib or in combination for 2 h, and protein expression was analyzed by Western blotting. Phosphorylation of ERK was inhibited by combination treatment (Fig 4A). To validate the apoptotic activity induced by drugs, the expression of the apoptosis marker cleaved by PARP was analyzed by Western blotting after treating H1993 CRH cells with crizotinib, trametinib, or in combination. The cleaved PARP expression was reduced with crizotinib and trametinib treatment but was increased in combination treatment (Fig 4B), indicating that the combination therapy was effective in crizotinib-resistant cells. We further analyzed the sensitivity to trametinib using a colony formation assay. The number of colonies was significantly reduced in combination therapy compared with crizotinib alone ($p = 0.034$), trametinib alone ($p = 0.044$), and control ($p = 0.017$; Fig 4C). These findings suggest that crizotinib resistance was addressed by combination therapy with trametinib and that MAPK pathway activation was an underlying mechanism for crizotinib resistance in the H1993 CRH cell line.

Using the cell viability assays, we explored the combinational effect between crizotinib and trametinib. Different concentrations of crizotinib and trametinib were administered to H1993 CRH, and growth inhibition percentage was calculated in a similar manner to the EBC-1 CRS. We then discovered the combination effect of these two drugs using the Chou–Talalay method for drug combination (Fig 4D). In H1993 CRH, the CI value of most combination points for crizotinib and trametinib was less than 1, revealing their synergistic effect.

## Phenotypic changes and EMT signatures upregulated in H1993 crizotinib-resistant cells

Following resistance to crizotinib, phenotypic changes in H1993 CRH cells were observed with the acquirement of spindle-shaped formation and elongation in the cellular structure compared with H1993 parental cells (Fig 1B). Furthermore, analyzing RNA sequence data by GSEA, the signatures related to EMT pathway expression in tumors were found to be enriched in the H1993 CRH cell line compared with parental cells, with an ES of 0.56 and a nominal $p$-value 0.0 (Fig 4E). Moreover, the expression level of mesenchymal markers such as N-cadherin, vimentin, and Zeb1 was upregulated at over 8.59-, 4.95-, and 3.16-fold in the H1993 CRH cell line compared with the expression of parental cells in accordance with the RNA sequence (Fig 4F). Thus, to confirm the abovementioned result, we explored this cell line by analyzing the Western blot expression of epithelial and mesenchymal markers, and these results are consistent with the observation in gene expression analysis (Fig 4G). The findings

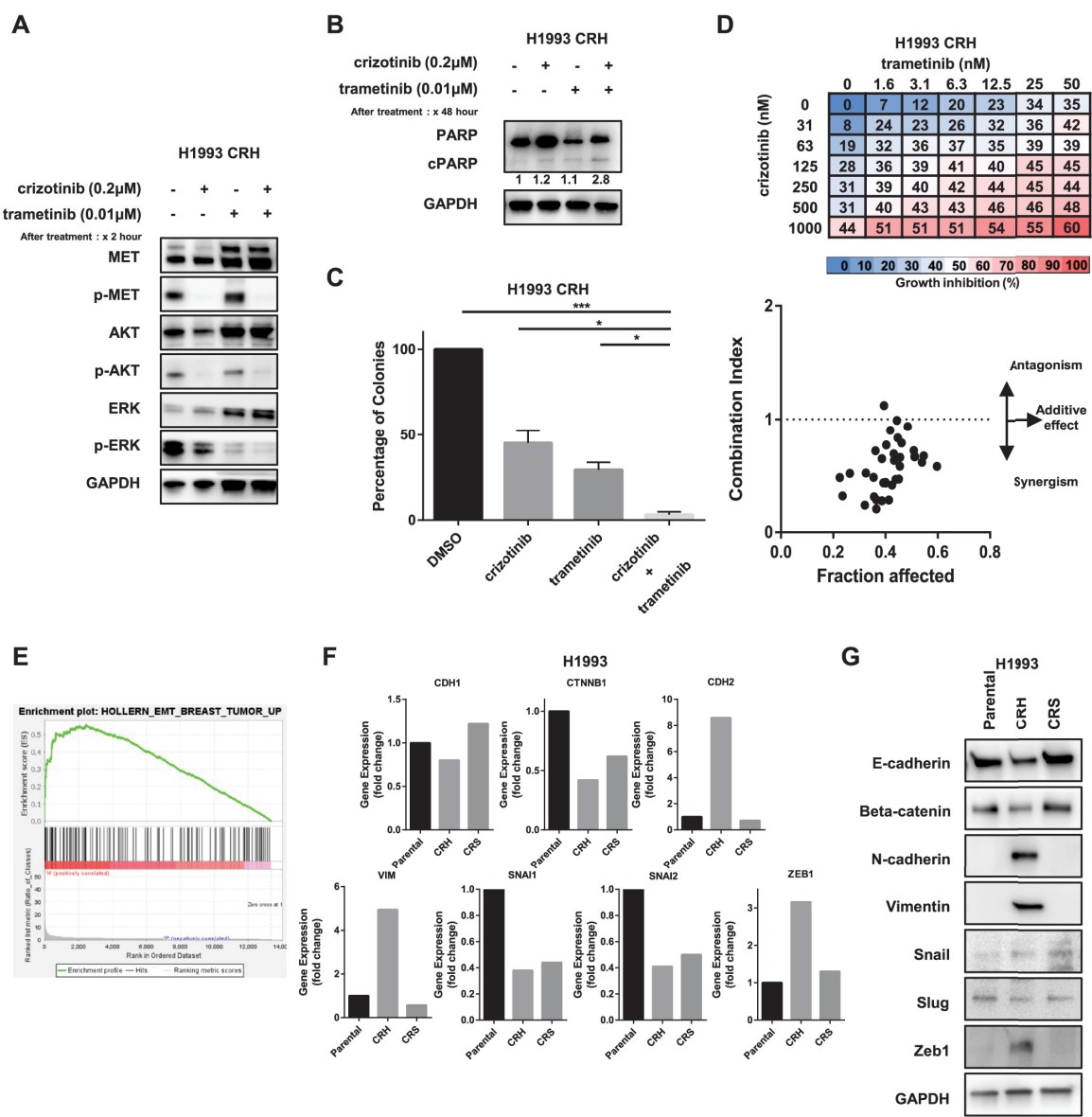

**Fig 4. MAPK pathway activation and EMT features are upregulated in the H1993 CRH cell line.** (A) H1993 CRH cells were treated with crizotinib (0.2 μM) and trametinib (0.1 μM) for 2 h, individually and in combination, and the lysates were subjected to immunoblotting. Expression of MET downstream signaling markers was detected. (B) The same cell line was treated with individual or a combination of crizotinib (0.2 μM) and trametinib (0.1 μM) for 48 h, and the lysate was subjected to immunoblotting. Apoptosis analysis of PARP and cleaved PARP expression in combination treatment with crizotinib and trametinib in H1993 CRH cells was observed by using Western blot data. (C) Combination treatment with crizotinib and trametinib inhibited colony formation in H1993 CRH cells. A colony formation assay was performed in H1993 CRH cells using a combination of crizotinib (0.2 μM) and trametinib (0.1 μM) individually or in combination treatment. Compared with DMSO (control), three independent assays were performed, and the average number of relative colonies formed was evaluated. (D) H1993 CRH was treated with a combination of crizotinib and trametinib for 96 h, and cell viability was determined. Data represented the mean value of the growth inhibition ratio at each concentration. The combination index (CI) plot indicates the CI and fraction affected (inhibition ratio) at different drug concentrations. (E) Gene set enrichment analysis (GSEA) in RNA sequence data of H1993 CRH cell lines compared with parental cell lines showing enrichment in the EMT-related gene set in RNA sequence data of H1993 CRH, with an enrichment score (ES) of 0.56 and a nominal *p*-value of 0.0. (F) Gene expression analysis of H1993 cells from RNA sequence data comparing genes encoding EMT markers in parental and resistant cell lines. The fold changes in each resistant cell line were calculated and compared with those in the parental cell line. (G) Western blot analysis of H1993 parental and resistant cells on EMT markers, which is consistent with the RNA sequence gene signature results. p- (Phosphorylated-), *$p < 0.05$, **$p < 0.01$, ***$p < 0.001$.

were suggestive of EMT as a mechanism underlying the acquired resistance to crizotinib in the H1993 CRH cell line.

## Discussion

Our study revealed several molecular mechanisms occurring in different cell lines with different approach backgrounds of acquired resistance to crizotinib. Interestingly, the crizotinib-resistant cell line established in our study showed resistance to crizotinib and other MET TKIs, such as tepotinib and cabozantinib. In the EBC-1 CRH cell line, we found that the overexpression of PAI-1 was associated with crizotinib resistance. The gene *SERPINE1* encodes the Serpin Family E Member 1 protein, which is also known as PAI-1, and this gene has been reported to be involved in tumor growth, angiogenesis, cancer cell survival, and metastasis through the regulation of several factors [22–24]. PAI-1 has been reported to confer resistance to various chemotherapeutic agents in cancer. High PAI-1 expression enhanced migration and apoptosis resistance, and such expression was associated with poor outcomes in head and neck carcinoma via the PI3K–AKT–mTOR pathway [25]. PAI-1 promotes colony formation and cell viability and decreases cisplatin-induced apoptosis via AKT-ERK signaling in esophageal squamous cell carcinoma [26]. PAI-1 also promotes actin cytoskeleton reorganization, glycolytic metabolism, migration and invasive phenotype, and orthotopic tumor growth via ERK signaling [27]. PAI-1 DNA promoter methylation is reported to be involved in carboplatin-induced EMT in epithelial ovarian cancer [28]. Angiogenesis is also known to be a mechanism of the effect of PAI-1 on tumorigenesis via vascular endothelial growth factors, such as the activation of the VEGFR-2 signaling pathway in gastric cancer and paclitaxel resistance in triple-negative cancer via VEGFA. Targeting PAI-1 inhibits angiogenesis, reduced cell proliferation, adhesion, colony formation, induction of apoptosis, and tumor growth in a human cancer xenograft model [29–31]. PAI-1 secreted via the autophagy pathway also contributes to chemotherapy resistance by modulating the tumor microenvironment [32]. Moreover, the high expression of the plasminogen activation signature and *SERPINE1* was associated with poor OS described within the TCGA lung adenocarcinoma RNA sequence dataset and clinical cohort [33]. We also showed that high PAI-1 expression was associated with poor OS in patients with lung cancer based on the Kaplan–Meier plotter database (Fig 2G). Furthermore, studies have revealed that PAI-1 inhibitors, developed initially as antithrombotic agents, were supportive as anticancer agents [31,34,35]. In our study, we used a small-molecule inhibitor of PAI-1, tiplaxtinin, along with crizotinib on resistant cells and revealed that inhibiting PAI-1 could reverse drug resistance. Consistent with our results, the knockdown of PAI-1 was also found to overcome drug resistance in other cancer types, such as triple-negative breast cancer and esophageal squamous cell carcinoma [26,28,30]. Therefore, the overexpression of PAI-1 was strongly associated with the mechanism of crizotinib resistance in the EBC-1 CRH cell line. Although the detailed mechanisms of this drug resistance remain unclear, previous studies have reported that apoptosis regulatory proteins, vascular endothelial growth factor (VEGF-A), SOX2, and AKT-ERK signaling pathway activation may contribute to drug resistance [24,26].

   The resistance mechanism in the EBC-1 CRS and H1993 cell lines have been elucidated as EMT and MAPK pathway activation driving drug resistance, with additional supportive evidence on antiapoptotic protein upregulation and extracellular matrix-related changes. In both of the EBC-1 CRS and H1993 CRH cells, we have revealed that the phosphorylated form of ERK was upregulated downstream of MET, thereby leading to the activation of cellular survival and proliferation. We further investigated the effect of the combination of crizotinib and the MEK inhibitor trametinib, where combining these two drugs overcomes sensitivity to

crizotinib. The MEK/ERK pathway is an important biological pathway in *MET*-amplified lung cancer [36] and a driver of therapy resistance in lung cancer, and this pathway is associated with poor survival [37]. The scope of this study revealed that the combination of MEK-targeted therapy along with crizotinib could be a potential therapeutic target.

Both of these cell lines showed EMT features as well as MAP kinase activation as a potential mechanism underlying crizotinib resistance. Although the mechanism underlying the role of EMT features and MAP kinase activation in crizotinib resistance in this cell line remains unclear, targeting EMT and/or MAP kinase using trametinib would be beneficial in these cells. Since the MAPK activation was found also in EBC-1 CRH according to Fig 1D, we also checked the effectiveness of MEK inhibitor therapy in this cell line. But the resistance to crizotinib was not overcome by trametinib in EBC-1 CRH.

EMT is a process by which epithelial cells change into mesenchymal-type cells. The loss of epithelial features causes tumor cells to escape the cytotoxic tumor microenvironment and favors cellular processes for tumor progression and proliferation. The acquisition of EMT features has been reported to be a possible mechanism underlying resistance to various drugs [38,39], and several EMT-related signaling pathways have been reported to be involved in drug resistance in cancer cells, such as the increase in drug efflux pumps and antiapoptotic effects [40]. Our study showed that EMT features were acquired during resistance to crizotinib, and all gene expression, protein expression, and GSEA of EMT gene set enrichment were strong suggestive evidence of acquiring resistance in this cell line.

In addition to EMT and MAP kinase activation in EBC-1 CRS, we also discovered the upregulation of the antiapoptotic protein Bcl-2 indicating the drug tolerance. The Bcl-2 family members of proteins include pro- and antiapoptotic proteins that act at the mitochondrial level and participate in cellular apoptosis. The inhibition of cell apoptosis by the upregulated antiapoptotic protein Bcl-2 confers a survival advantage, and its inhibition has been known to be involved in chemotherapeutic resistance in various types of cancers [41,42]. Although the exact molecular mechanism of the involvement of EMT in drug resistance remains unclear, Bcl-2 has also been reported to be associated with EMT by *CDH1* inactivation, BCL-2/TWIST1 complex formation, and its upregulation inducing EMT through the N-cadherin/FGFR/extracellular signal-regulated kinase pathways [42–44]. Our study discovered an important association between the antiapoptotic upregulation and EMT co-regulation of the drug resistance mechanism.

Extracellular matrices (ECMs) are multicomponent networks that surround cells in tissues and serve as scaffolds for tissue organization, and ECMs are necessary for cell survival, growth, and differentiation. ECM remodeling has a wide range of pathological effects during carcinogenesis, and it modulates the hallmarks of cancer in many aspects, including resisting cell death and promoting genomic instability [45,46]. Various ECM proteins, such as fibronectin and laminin, can activate signaling pathways, including integrin-mediated signaling, which triggers EMT-related transcription factors, and promote EMT. Consequently, EMT alters ECM mechanics and composition, including matrix metalloproteinase-driven tissue remodeling in the extracellular matrix [46,47]. In our study, we discovered that the ECM-related gene signatures are enriched in the EBC-1 CRS cell line, revealing that ECM signatures are important in molecular changes that lead to EMT and greatly impact therapeutic resistance.

However, we could not find any relevant resistance mechanism underlying the H1993 CRS cell line in either genomic or transcriptomic analyses, including whole-exome sequencing. No definite pathogenic changes were identified in this cell line's mutational profiles. The resistance mechanism remains controversial, and further investigations are recommended.

Therefore, we established four cell lines from two parental *MET*-amplified lung carcinoma cell lines, which have acquired resistance to crizotinib and revealed various possible molecular

**Table 1. Summary of the mechanisms of crizotinib resistance in MET NSCLC.**

| Cell Line | Mechanism of Resistance and features | Suggestions |
|---|---|---|
| EBC1 CRH | Upregulation of PAI-1 | Combination with PAI-1 inhibitor |
| EBC1 CRS | Down stream MAP kinase pathway activation and epithelial to mesenchymal transition with extracellular matrix reorganization and upregulation of anti-apoptotic protein | Combination with MEK inhibitor |
| H1993 CRH | Down stream MAP kinase pathway activation and epithelial to mesenchymal transition | Combination with MEK inhibitor |
| H1993 CRS | No relevant finding | |

mechanisms of resistance from PAI-1 overexpression, involvement of MAPK pathway activation and acquisition of EMT features to possible therapeutic strategies to address the resistance using combination therapy with various therapeutic agents such as tiplaxtinin and trametinib (Table 1). We identified PAI-1 overexpression as a novel therapeutic target for crizotinib-resistant MET NSCLC which provides rationale for the future clinical evaluation of its combination approach for patients with MET NSCLC. Further exploration of the research findings in clinical trials is needed to assess the safety and efficacy in real-world clinical settings. Taken together, findings from this study would be of great importance in overcoming drug resistance after treatment and in the choice of personalized medicine for cancer treatment.

## Supporting information

**S1 Fig. Original uncropped western blot images.**
(PDF)

**S2 Fig. Knockdown of PAI-1 by transfection with shRNA.** Control cell lines were treated with shNT (nontarget), and two knockdown cell lines were established and validated by assessing the mRNA gene expression of *SERPINE1*.
(EPS)

**S3 Fig. EMT- and ECM-related gene set enrichment in GSEA in curated (C2all) EBC-1 CRS cell line gene sets.**
(EPS)

**S4 Fig. DAVID functional GO analysis of gene enrichment in the EBC-1 CRS cell line.** Enrichment of genes related to the extracellular matrix in biological processes (S4A) and molecular functions (S4B).
(EPS)

## Acknowledgments

We thank Ms. Fumiko Isobe (General Thoracic Surgery and Breast and Endocrinological Surgery Department, Graduate School of Medicine, Dentistry and Pharmaceutical Sciences, Okayama University, Okayama, Japan) for her technical assistance.

## Author Contributions

**Conceptualization:** Yin Min Thu, Ken Suzawa, Keiichi Date, Shinichi Toyooka.

**Data curation:** Yin Min Thu, Shuta Tomida, Kosuke Ochi.

**Formal analysis:** Yin Min Thu, Shuta Tomida, Kosuke Ochi, Shimpei Tsudaka, Fumiaki Takatsu, Keiichi Date, Naoki Matsuda, Kazuma Iwata, Kentaro Nakata, Kazuhiko Shien, Hiromasa Yamamoto, Mikio Okazaki, Seiichiro Sugimoto, Shinichi Toyooka.

**Funding acquisition:** Ken Suzawa, Keiichi Date, Shinichi Toyooka.

**Investigation:** Yin Min Thu, Kosuke Ochi.

**Methodology:** Yin Min Thu, Kosuke Ochi.

**Project administration:** Ken Suzawa, Shuta Tomida, Shinichi Toyooka.

**Resources:** Ken Suzawa.

**Software:** Shuta Tomida.

**Supervision:** Ken Suzawa, Keiichi Date, Kazuhiko Shien, Hiromasa Yamamoto, Shinichi Toyooka.

**Visualization:** Shuta Tomida.

**Writing – original draft:** Yin Min Thu.

**Writing – review & editing:** Ken Suzawa, Shuta Tomida, Kosuke Ochi, Shimpei Tsudaka, Fumiaki Takatsu, Keiichi Date, Naoki Matsuda, Kazuma Iwata, Kentaro Nakata, Kazuhiko Shien, Hiromasa Yamamoto, Mikio Okazaki, Seiichiro Sugimoto, Shinichi Toyooka.

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
