## [Decision Letter · Decision Letter 0]

16 Jan 2024

PONE-D-23-42914PAI-1 mediates resistance to MET-targeted therapy in non-small cell lung cancerPLOS ONE

Dear Dr. Suzawa,

Thank you for submitting your manuscript to PLOS ONE. After careful consideration, we feel that it has merit but does not fully meet PLOS ONE’s publication criteria as it currently stands. Therefore, we invite you to submit a revised version of the manuscript that addresses the points raised during the review process.

We look forward to receiving your revised manuscript.

Kind regards,

Abeer El Wakil, PhD

Academic Editor

PLOS ONE

Journal Requirements:

“This work was supported by Japan Society for the Promotion of Science (JSPS) KAKENHI Grant Number JP19K18216.”

3. Thank you for providing the following Funding Statement: 

“I have read the journal's policy and the authors of this manuscript have the following competing interests: Shinichi Toyooka received research funding from Eli Lilly Japan, Taiho (Japan) and Chugai (Japan), and lecture fees from Chugai. All other authors have declared that no competing interests exist.”

We note that one or more of the authors is affiliated with the funding organization, indicating the funder may have had some role in the design, data collection, analysis or preparation of your manuscript for publication; in other words, the funder played an indirect role through the participation of the co-authors.

If the funding organization did not play a role in the study design, data collection and analysis, decision to publish, or preparation of the manuscript and only provided financial support in the form of authors' salaries and/or research materials, please review your statements relating to the author contributions, and ensure you have specifically and accurately indicated the role(s) that these authors had in your study in the Author Contributions section of the online submission form. Please make any necessary amendments directly within this section of the online submission form.  Please also update your Funding Statement to include the following statement: “The funder provided support in the form of salaries for authors [insert relevant initials], but did not have any additional role in the study design, data collection and analysis, decision to publish, or preparation of the manuscript. The specific roles of these authors are articulated in the ‘author contributions’ section.”

If the funding organization did have an additional role, please state and explain that role within your Funding Statement.

Please also provide an updated Competing Interests Statement declaring this commercial affiliation along with any other relevant declarations relating to employment, consultancy, patents, products in development, or marketed products, etc. 

Additional Editor Comments (if provided):

While the study investigated resistance mechanisms to MET tyrosine kinase inhibitors in MET-amplified lung cancer cell lines and addressed an important clinical challenge, it presents several significant limitations that undermine its scientific significance and clinical relevance. The reviewers have raised several concerns that need to be addressed before the manuscript could be accepted for publication.

Reviewers' comments:

Reviewer's Responses to Questions

**Comments to the Author**

1. Is the manuscript technically sound, and do the data support the conclusions?

Reviewer #1: No

Reviewer #2: Yes

Reviewer #3: Partly

Reviewer #4: Yes

2. Has the statistical analysis been performed appropriately and rigorously? 

Reviewer #1: No

Reviewer #2: Yes

Reviewer #3: No

Reviewer #4: Yes

3. Have the authors made all data underlying the findings in their manuscript fully available?

Reviewer #1: No

Reviewer #2: Yes

Reviewer #3: No

Reviewer #4: Yes

4. Is the manuscript presented in an intelligible fashion and written in standard English?

Reviewer #1: No

Reviewer #2: Yes

Reviewer #3: No

Reviewer #4: Yes

5. Review Comments to the Author

Reviewer #1: The study investigating resistance mechanisms to MET tyrosine kinase inhibitors in MET-amplified lung cancer cell lines, while addressing an important clinical challenge, presents several significant limitations that undermine its scientific significance and clinical relevance:

1. The study focuses on only two lung cancer cell lines, EBC-1 and H1993. This narrow scope may not adequately represent the genetic and phenotypic heterogeneity observed in non-small cell lung cancer (NSCLC) patients. Broader inclusion of cell lines could provide a more comprehensive understanding of resistance mechanisms.

2. The absence of in vivo studies, such as patient-derived xenograft models, limits the validation of the proposed resistance mechanisms and their therapeutic implications. In vivo studies are crucial to confirm the relevance of in vitro findings in a more complex biological context.

3. While the study identifies SERPINE1 overexpression and epithelial-to-mesenchymal transition in EBC-1 cells and MAP kinase pathway activation in H1993 cells as resistance mechanisms, it may overlook other potentially significant pathways. A more comprehensive approach could provide a fuller picture of resistance dynamics.

4. The study does not include clinical data or samples to correlate the in vitro findings with patient outcomes. Clinical validation is essential to confirm the relevance of the identified resistance mechanisms in actual patient populations.

5. Although the study suggests combination therapy with a PAI-1 inhibitor and MEK inhibitors as potential strategies, these suggestions are not experimentally validated within the study. More experimental testing of combination therapies is necessary to assess their efficacy and safety.

6. The study does not explore the development of biomarkers for predicting or monitoring resistance to MET-TKIs, which could be crucial for personalized therapy in NSCLC.

In summary, while the study contributes to understanding resistance mechanisms to MET-TKIs in MET-amplified lung carcinoma, its limited scope, reliance on in vitro models, lack of in vivo validation, and absence of clinical correlation significantly restrict its impact on the field. Further research involving a broader range of cell lines, in vivo studies, and clinical validation is required to enhance the translational potential of these findings.

Reviewer #2: 1. Title: "PAI-1 as resistance..." - Can you precise it as "acquired resistance" or do you have any signals that it also can be intrinsic? Furthermore, the title address only one resistance mechanism while the whole article dedicates also two other, EMT driven and via ERK upregulation. Please consider re-phrasing the title.

2. line 44: add "often" to "mutually exclusive"

3. Line 77: "second site mutation" should be changed to "off-target resistance"

4. Line 85: you have used cell lines both from squamous and adenocarcinoma. Please, explain the reason and address the issue of possible different resistance mechanism depending of the primary phenotype. Is the type of phenotype predisposing to a certain resistance. What was the level of MET amplification, as it a priori defines response to Crizotinib.

5. Line 148. please provide reference for the human genome (hg38)

6. Discussion: The capital should be enriched by more discussion about validation of PAI-overexpression and clinical implementation. What is the next steps for employing your finding into the clinic? Proposal of clinical trials?

Line 513: Neither Tiplaxatin or Trametinib are chemotherapeutic agents.

To provide more focused and comprehensive insight in your findings, a table collecting possible resistance mechanism, their features and suggestion for overcoming resistance will be an advantage for readers.

Reviewer #3: Major concern

1. Since all the cell viability assays are performed with six replicates, data for cell viability shown in Fig 1A need to be expressed with a standard deviation.

2. Signal transduction responses, shown in fig 1C, between two resistant-induced cell lines are different. The authors need to discuss the fundamental differences between these two induction processes and how they reflect on the clinical outcomes of patients.

3. The authors described that the SERPINE1 gene is six-fold higher in EBC-1 CRH cells. Please explain the disproportionate cell viability response in combination with MET-TKI and PAI-1 inhibitors (fig 1C).

4. There are no significant morphology differences between CRH and CRS in Fig 1B but substantial differences in the EMT-marker protein expression in Fig 3 and Fig 4. Please explain it.

5. It is not appropriate to conclude that EMT-induced antiapoptotic Bcl-2 expression in line 321-324 without any data support.

6. The authors need to conduct all the Western analyses of the EMT transition marker proteins with significant expression differences in gene analysis.

Reviewer #4: This manuscript reports the identification of mechanisms underlying crizotinib resistance in two MET-amplified lung cancer cell lines (EBC-1 and H1993). The authors demonstrated that multiple factors, such as PAI-1 overexpression and activation of EMT and MAPK pathways, were involved in the crizotinib resistance mechanism in these cell lines. The findings provide insights into potential therapeutic strategies for crizotinib-resistant MET-amplified lung cancer.

I would like to have some issues to be addressed by the authors.

1. One of the MET-amplified lung cancer cell lines used in this study was squamous cell carcinoma and the other was adenocarcinoma (page 5, line 85). In their introduction, the authors noted that MET amplification accounts for 1.7-2.5% of lung adenocarcinomas, but they did not mention MET amplification in squamous cell carcinoma. The authors should explain why they used squamous cell carcinoma cell lines for this study. Also, what is the percentage of MET amplification in lung squamous cell carcinoma?

2. The p-ERK activation has been observed in EBC-1 CRH, EBC-1 CRS, and H1993 CRH cell lines (page 11, line 218). Have you examined whether the MEK inhibitor combination is also effective in EBC-1 CRH and EBC-1 CRS?

3. As mentioned above, EBC-1 is a lung squamous cell carcinoma cell line, and it would be desirable to explain whether this finding that PAI-1 is involved in crizotinib resistance in EBC-1 CRH can be extrapolated to adenocarcinoma as well (page 13, line 24). (page 13, line 242).

4. Please describe how you titrated the concentration of tiplaxtinin (page 13, line 255).

5. Please describe which data were used for the analysis of survival data of NSCLC patients and the method of testing in the method section (page 14, line 268). In addition, since this study was focused on MET-amplified lung cancer patients, it may be better to consider conducting the survival analysis using only MET-amplified lung cancer patients.

6. What is the possible interpretation of the fact that only SNAI1 was not changed in CRS but up-regulated in CRH in the gene expression analysis of EMT markers in EBC-1 CRS cell lines (page 16, line 304)?

7. In the western blotting of EMT-related proteins (Fig. 3C), the authors stated that N-cadherin and vimentin are highly expressed in EBC-1 CRS, but this does not seem to be the case in the images (especially vimentin) (page 16, line 309). There does not seem to be a significant change compared to CRH.

8. H993 seems to be a misnomer for H1933 (page 19, line 354).

9. Cleaved PARP expression in H1993 CRH cell lines does not appear to be significantly different between crizotinib, trametinib, and combination treatment groups (page 19, line 355). Have you considered quantification or evaluation by other apoptosis markers?

10. What is the possible interpretation of the lower expression of SNAI1 and SNAI2 in H1993 CRH cell lines compared to parental cells (page 21, line 382)?

11. Regarding MET in lung cancer, treatment targeting MET exon 14 skipping mutation has been attracting much attention. How do the authors think the results of this study can be applied to MET ex14 lung cancers?

6. PLOS authors have the option to publish the peer review history of their article (what does this mean?). If published, this will include your full peer review and any attached files.

Reviewer #1: No

Reviewer #2: No

Reviewer #3: **Yes: **Jun Jen Liu

Reviewer #4: No

---

## [Author Response · Author response to Decision Letter 0]

23 Feb 2024

We are grateful to the reviewers for their thoughtful comments and questions. The following are our responses. We have amended the manuscript to where appropriate to these changes.

Reviewer #1: 

1. The study focuses on only two lung cancer cell lines, EBC-1 and H1993. This narrow scope may not adequately represent the genetic and phenotypic heterogeneity observed in non-small cell lung cancer (NSCLC) patients. Broader inclusion of cell lines could provide a more comprehensive understanding of resistance mechanisms.

Response:

We agree with reviewer#1 that the narrow scope of genetic and phenotypic heterogeneity should be considered in representing NSCLC patients. However, in this study, we mainly focus on MET-dependent NSCLC and therefore, we are using each of the adenocarcinoma and squamous cell carcinoma MET-amplified cell lines. We are currently conducting a further study on the PAI-1 effect in EGFR-, KRAS-, and ALK-dependent cell lines, however, the data are still immature and could not be included in this paper.

2. The absence of in vivo studies, such as patient-derived xenograft models, limits the validation of the proposed resistance mechanisms and their therapeutic implications. In vivo studies are crucial to confirm the relevance of in vitro findings in a more complex biological context.

Response:

We thank reviewer#1 for the valuable comment. Regarding in-vivo studies, we did not have a chance to get patient-derived xenograft samples from patients with MET NSCLC. During the preliminary experiments, the EBC-1 CRH cell line did not form xenograft tumors. Therefore, we could not include the in-vivo data in this study.

3. While the study identifies SERPINE1 overexpression and epithelial-to-mesenchymal transition in EBC-1 cells and MAP kinase pathway activation in H1993 cells as resistance mechanisms, it may overlook other potentially significant pathways. A more comprehensive approach could provide a fuller picture of resistance dynamics.

Response:

As reviewer#1 says, we cannot deny the possibilities of other resistance mechanisms. In this study, we performed the RNA Sequence and whole exome sequence (WES) as a comprehensive screening to find the candidates for the resistance. As far as we have analyzed the data, we could not identify other candidates other than PAI-1 and MAP kinase pathway as resistance mechanisms. Notably, inhibiting the PAI-1 or MAP kinase pathway restores the resistance partly. We believe that this means that these resistance mechanisms contribute to the resistance.

4. The study does not include clinical data or samples to correlate the in vitro findings with patient outcomes. Clinical validation is essential to confirm the relevance of the identified resistance mechanisms in actual patient populations.

Response:

We agree with reviewer#1 that clinical validation is essential to confirm the relevance. Despite the crucial role of clinical data or samples, we did not have a chance to get the patients’ samples with MET NSCLC as mentioned before. Therefore, we could not conclude clinical data in this study.

5. Although the study suggests combination therapy with a PAI-1 inhibitor and MEK inhibitors as potential strategies, these suggestions are not experimentally validated within the study. More experimental testing of combination therapies is necessary to assess their efficacy and safety.

Response:

Thank you very much for reviewer#1’s constructive advice. Regarding experimental validation, we have performed the in-vitro experiments to determine the efficacy of the combination treatments (Figures 2C and 2D for PAI-1 and Figures 3A, 3B, 3C, 3D, 4A, 4B, 4C and 4D for MEK inhibitor). However, as we mentioned before, we could not conduct the in-vivo experiments for further assessment of their efficacy and safety, which is the limitation of this study.

6. The study does not explore the development of biomarkers for predicting or monitoring resistance to MET-TKIs, which could be crucial for personalized therapy in NSCLC.

Response:

We thank reviewer#1 for the comment. Our study revealed that the overexpression of PAI-1 contributes to the drug resistance to MET-TKI, indicating that PAI-1 could be a biomarker as well as a therapeutic target for resistance to MET-TKI. 

Reviewer #2: 

1. Title: "PAI-1 as resistance..." - Can you precise it as "acquired resistance" or do you have any signals that it also can be intrinsic? 

Furthermore, the title addresses only one resistance mechanism while the whole article dedicates also two other, EMT-driven and via ERK upregulation. Please consider re-phrasing the title.

Response:

We thank reviewer#2 for the detailed review of our manuscript and the thoughtful suggestion regarding rephrasing the title. We agree to include “PAI-1 as acquired resistance”. Since we discovered PAI-1 as the strongest novelty in elucidating the crizotinib resistance in MET NSCLC, we would like to emphasize our novel finding. 

2. line 44: add "often" to "mutually exclusive"

Response:

We thank reviewer#2 for the correction. We have amended in the manuscript (Page 3, Line 49).

3. Line 77: "second site mutation" should be changed to "off-target resistance"

Response:

We thank reviewer#2 for pointing this out. We think the second site mutation within the MET should be “on-target resistance” and therefore, we changed the phrasing as “on-target resistance in the kinase domain and off-target resistance by bypass signaling pathway activation” in our manuscript (Page 4, Line 79).

4. Line 85: you have used cell lines both from squamous and adenocarcinoma. Please, explain the reason and address the issue of possible different resistance mechanisms depending of the primary phenotype. Is the type of phenotype predisposing to a certain resistance. What was the level of MET amplification, as it a priori defines response to Crizotinib.

Response:

In this study, we would like to mention the MET-dependent NSCLC cell lines and these two cell lines, EBC-1 and H1993, are the major commercially available MET-amplified lung cancer cell lines. Our results showed different mechanisms of resistance among the cell lines, but we could not conclude that histology determines the resistance mechanism because we only used two cell lines.

Regarding the level of MET amplification, we added the data on the MET copy number in the cell lines we used in this study (Figure 1C).

5. Line 148. please provide reference for the human genome (hg38)

Response:

Thank you for pointing this out. We have added the reference for the human genome (hg38) in the manuscript (Page 10, Line 179).

https://www.ncbi.nlm.nih.gov/datasets/genome/GCF_000001405.37/

6. Discussion: The capital should be enriched by more discussion about validation of PAI-overexpression and clinical implementation. What is the next steps for employing your finding into the clinic? Proposal of clinical trials?

Response:

We are thankful for the valuable comment from reviewer#2. We have added the possible next steps for employing our findings in the clinics and proposal of clinical trials in the discussion section of the manuscript as follows (Page 33, Line 607). “We identified PAI-1 overexpression as a novel therapeutic target for crizotinib-resistant MET NSCLC which provides rationale for the future clinical evaluation of its combination approach for patients with MET NSCLC. Further exploration of the research findings in clinical trials is needed to assess the safety and efficacy in real-world clinical settings.”

7. Line 513: Neither Tiplaxatin or Trametinib are chemotherapeutic agents.

Response:

We again thank reviewer#2 for the correction. We have amended it in the manuscript (Page 33, Line 606).

8. To provide more focused and comprehensive insight in your findings, a table collecting possible resistance mechanisms, their features and suggestions for overcoming resistance will be an advantage for readers.

Response:

We thank reviewer#2 for the great suggestions and comments. For more focus and comprehensive insight, we added the Table 1 in the manuscript (Page 34, Line 614).

Reviewer #3: 

1. Since all the cell viability assays are performed with six replicates, data for cell viability shown in Fig 1A need to be expressed with a standard deviation.

Response:

We would like to thank reviewer#3 for the great advice. We have added the necessary data in the amended manuscript (figure 1A and Page 12, Line 234).

2. Signal transduction responses, shown in fig 1C, between two resistant-induced cell lines are different. The authors need to discuss the fundamental differences between these two induction processes and how they reflect on the clinical outcomes of patients.

Response:

We thank reviewer#3 for the great suggestion. We have added the detailed method in the establishment of two resistant cell lines in the materials and method section (

 6, Line 96). In the actual clinical courses, the resistance mechanisms is complicated and heterogeneous. The standard method to establish resistance cells in-vitro is chronic exposure to escalating dose of therapeutic drug. But in clinical setting, patients are treated with maximum dose from the start of treatment. We think there is a discrepancy between the two. So, we originally established the high-dose method, exposure to maximum dose of drug from the beginning, because we think that this method would reflect the actual clinical practice. According to these reasons, we used both methods in establishing the resistant cells.

3. The authors described that the SERPINE1 gene is six-fold higher in EBC-1 CRH cells. Please explain the disproportionate cell viability response in combination with MET-TKI and PAI-1 inhibitors (fig 1C).

Response:

Since reviewer #3 mentions the SERPINE1 gene, we think reviewer#3 is mentioning Figure 2. Since inhibiting PAI-1 with its inhibitor tiplaxtinin in combination with MET-TKI crizotinib restores the resistance to crizotinib (Figure 2C and 2D), we think that this represents the PAI-1 overexpression in this EBC-1 CRH cell line contributes to or mediate the drug resistance.

4. There are no significant morphology differences between CRH and CRS in Fig 1B but substantial differences in the EMT-marker protein expression in Fig 3 and Fig 4. Please explain it.

Response:

We thank reviewer#3 for the comment. EMT status of the cells is regulated by a variety of pathways and multiple factors are affecting the expression. We are thinking that EMT markers often show different expression patterns of each gene representing EMT markers.

5. It is not appropriate to conclude that EMT-induced antiapoptotic Bcl-2 expression in line 321-324 without any data support.

Response:

We thank reviewer#3 for constructive comment. We omitted this description in the result section and only discussed it in the discussion section of our manuscript.

6. The authors need to conduct all the Western analyses of the EMT transition marker proteins with significant expression differences in gene analysis.

Response:

According to reviewer#3’s thoughtful suggestions, we conducted all the western blot analyses of the EMT marker proteins we mentioned in the gene expression analysis (Figures 3G and 4G). We found the expression of other EMT-related proteins is in concordance with the gene expression findings.

Reviewer #4: 

1. One of the MET-amplified lung cancer cell lines used in this study was squamous cell carcinoma and the other was adenocarcinoma (page 5, line 85). In their introduction, the authors noted that MET amplification accounts for 1.7-2.5% of lung adenocarcinomas, but they did not mention MET amplification in squamous cell carcinoma. The authors should explain why they used squamous cell carcinoma cell lines for this study. Also, what is the percentage of MET amplification in lung squamous cell carcinoma?

Response:

We thank reviewer#4 for the comments and questions. We aimed to elucidate the drug resistance mechanism in MET-dependent NSCLC cell lines

and EBC-1 and H1993, are the major commercially available MET-amplified lung cancer cell lines. Our results showed different mechanisms of resistance among the cell lines, but we could not conclude that histology determines the resistance mechanism because we only used two cell lines. On account for the percentage of MET amplification in lung squamous cell carcinoma, according to cBioportal database (https://www.cbioportal.org/), 0.013% was found to have MET amplified in lung squamous cell carcinoma.

2. The p-ERK activation has been observed in EBC-1 CRH, EBC-1 CRS, and H1993 CRH cell lines (page 11, line 218). Have you examined whether the MEK inhibitor combination is also effective in EBC-1 CRH and EBC-1 CRS?

Response:

We are grateful for this insightful question. We examined the effectiveness of MEK inhibitor combination treatment in EBC-1 CRH and EBC-1 CRS as reviewer#4 suggested. We found out that the MEK inhibitor combination treatment is effective in EBC-1 CRS, but not in EBC-1 CRH. We added these new findings of EBC-1 CRS in the amended manuscript result section (Figures 3A, 3B, 3C and 3D) (Page 18, Line 342).

3. As mentioned above, EBC-1 is a lung squamous cell carcinoma cell line, and it would be desirable to explain whether this finding that PAI-1 is involved in crizotinib resistance in EBC-1 CRH can be extrapolated to adenocarcinoma as well (page 13, line 24). (page 13, line 242).

Response:

We thank reviewer#4 for the valuable comment. Our data is not enough to conclude this. We could only identify the PAI-1 mediated crizotinib resistance mechanism restricted to squamous cell carcinoma cell line (EBC-1 CRH) so far.

4. Please describe how you titrated the concentration of tiplaxtinin (page 13, line 255).

Response:

We used the tiplaxtinin concentration of 10�M in our experiments. We determined the concentration which had a mild inhibitory effect on the cells, and we also referred to the previous reports. [1, 2]

5. Please describe which data were used for the analysis of survival data of NSCLC patients and the method of testing in the method section (page 14, line 268). In addition, since this study was focused on MET-amplified lung cancer patients, it may be better to consider conducting the survival analysis using only MET-amplified lung cancer patients.

Response:

We added the updated description of survival analysis data of Lung cancer patients according to the Kaplan-Meier plotter in our material and methods section (Page 11, Line 219). We agree with reviewer#4 that it is better to conduct survival analysis using only MET-amplified lung cancer patients. However, since we do not have the genomic profile of MET-amplified lung cancer patients, we are unable to conduct such an analysis. 

6. What is the possible interpretation of the fact that only SNAI1 was not changed in CRS but up-regulated in CRH in the gene expression analysis of EMT markers in EBC-1 CRS cell lines (page 16, line 304)?

Response:

We thank reviewer#4 for the comment. Since EMT status of the cells is regulated by a variety of pathways and multiple factors are affecting the expression, EMT markers often show different expression patterns of each gene representing EMT markers. We could not clearly explain the exact mechanism of this heterogeneity in expression of EMT markers so far.

7. In the western blotting of EMT-related proteins (Fig. 3C), the authors stated that N-cadherin and vimentin are highly expressed in EBC-1 CRS, but this does not seem to be the case in the images (especially vimentin) (page 16, line 309). There does not seem to be a significant change compared to CRH.

Response:

We agree with reviewer#4 in that the expressions of EMT-related protein markers are slightly controversial. We observed the reduced expression of beta-catenin and increased expression of N-cadherin and Slug in this analysis. As we mentioned above, since their expressions are influenced by various pathways, the expression patterns differ greatly among the markers. we have amended the manuscript to describe a little more detail. 

8. H993 seems to be a misnomer for H1933 (page 19, line 354).

Response:

We thank reviewer#4 for the correc

---

## [Decision Letter · Decision Letter 1]

4 Mar 2024

PAI-1 mediates acquired resistance to MET-targeted therapy in non-small cell lung cancer

PONE-D-23-42914R1

Dear Dr. Suzawa,

We’re pleased to inform you that your manuscript has been judged scientifically suitable for publication and will be formally accepted for publication once it meets all outstanding technical requirements.

Kind regards,

Abeer El Wakil, PhD

Academic Editor

PLOS ONE

Additional Editor Comments (optional):

Reviewers' comments:

Reviewer's Responses to Questions

**Comments to the Author**

1. If the authors have adequately addressed your comments raised in a previous round of review and you feel that this manuscript is now acceptable for publication, you may indicate that here to bypass the “Comments to the Author” section, enter your conflict of interest statement in the “Confidential to Editor” section, and submit your "Accept" recommendation.

Reviewer #4: All comments have been addressed

2. Is the manuscript technically sound, and do the data support the conclusions?

Reviewer #4: Yes

3. Has the statistical analysis been performed appropriately and rigorously? 

Reviewer #4: Yes

4. Have the authors made all data underlying the findings in their manuscript fully available?

Reviewer #4: Yes

5. Is the manuscript presented in an intelligible fashion and written in standard English?

Reviewer #4: Yes

6. Review Comments to the Author

Reviewer #4: The authors successfully addressed my comments and I would be satisfied with the authors' response.

7. PLOS authors have the option to publish the peer review history of their article (what does this mean?). If published, this will include your full peer review and any attached files.

Reviewer #4: No

---

## [Editor Report · Acceptance letter]

8 May 2024

PONE-D-23-42914R1 

PLOS ONE

Dear Dr. Suzawa, 

I'm pleased to inform you that your manuscript has been deemed suitable for publication in PLOS ONE. Congratulations! Your manuscript is now being handed over to our production team.

Kind regards, 

on behalf of

Professor Abeer El Wakil 

Academic Editor

PLOS ONE